# Alternating minimization for dictionary learning with random initialization

**Niladri S. Chatterji**
University of California, Berkeley
chatterji@berkeley.edu

**Peter L. Bartlett**
University of California, Berkeley
peter@berkeley.edu

## Abstract

We present theoretical guarantees for an alternating minimization algorithm for the dictionary learning/sparse coding problem. The dictionary learning problem is to factorize vector samples $y^1, y^2, \ldots, y^n$ into an appropriate basis (dictionary) $A^*$ and sparse vectors $x^{1*}, \ldots, x^{n*}$. Our algorithm is a simple alternating minimization procedure that switches between $\ell_1$ minimization and gradient descent in alternate steps. Dictionary learning and specifically alternating minimization algorithms for dictionary learning are well studied both theoretically and empirically. However, in contrast to previous theoretical analyses for this problem, we replace a condition on the operator norm (that is, the largest magnitude singular value) of the true underlying dictionary $A^*$ with a condition on the matrix infinity norm (that is, the largest magnitude term). Our guarantees are under a reasonable generative model that allows for dictionaries with growing operator norms, and can handle an arbitrary level of overcompleteness, while having sparsity that is information theoretically optimal. We also establish upper bounds on the sample complexity of our algorithm.

*Erratum, August 7, 2019: An earlier version of this paper appeared in NIPS 2017 which had an erroneous claim about convergence guarantees with random initialization. The main result – Theorem 3 – has been corrected by adding an assumption about the initialization (Assumption B1).*

## 1 Introduction

In the problem of sparse coding/dictionary learning, given i.i.d. samples $y^1, y^2, \ldots, y^n \in \mathbb{R}^d$ produced from the generative model

$$y^i = A^* x^{i*} \tag{1}$$

for $i \in \{1, 2, \ldots, n\}$, the goal is to recover a fixed dictionary $A^* \in \mathbb{R}^{d \times r}$ and $s$-sparse vectors $x^{i*} \in \mathbb{R}^r$. (An $s$-sparse vector has no more than $s$ non-zero entries.) In many problems of interest, the dictionary is often overcomplete, that is, $r \geq d$. This is believed to add flexibility in modeling and robustness. This model was first proposed in neuroscience as an energy minimization heuristic that reproduces features of the V1 region of the visual cortex (Olshausen and Field, 1997; Lewicki and Sejnowski, 2000). It has also been an extremely successful approach to identifying low dimensional structure in high dimensional data; it is used extensively to find features in images, speech and video (see, for example, references in Elad and Aharon, 2006).

Most formulations of dictionary learning tend to yield non-convex optimization problems. For example, note that if either $x^{i*}$ or $A^*$ were known, given $y^i$, this would just be a (matrix/sparse) regression problem. However, estimating both $x^{i*}$ and $A^*$ simultaneously leads to both computational as well as statistical complications. The heuristic of alternating minimization works well empirically for dictionary learning. At each step, first an estimate of the dictionary is held fixed while the sparse

coefficients are estimated; next, using these sparse coefficients the dictionary is updated. Note that in *each step* the sub-problem has a convex formulation, and there is a range of efficient algorithms that can be used. This heuristic has been very successful empirically, and there has also been significant recent theoretical progress in understanding its performance, which we discuss next.

## 1.1 Related Work

A recent line of work theoretically analyzes local linear convergence rates for alternating minimization procedures applied to dictionary learning (Agarwal et al., 2014; Arora et al., 2015). Arora et al. (2015) present a neurally plausible algorithm that recovers the dictionary exactly for sparsity up to $s = \mathcal{O}(\sqrt{d}/(\mu \log(d)))$, where $\mu/\sqrt{d}$ is the level of incoherence in the dictionary (which is a measure of the similarity of the columns; see Assumption A1 below). Agarwal et al. (2014) analyze a least squares/$\ell_1$ minimization scheme and show that it can tolerate sparsity up to $s = \mathcal{O}(d^{1/6})$. Both of these establish local linear convergence guarantees for the maximum column-wise distance. Exact recovery guarantees require a singular-value decomposition (SVD) or clustering based procedure to initialize their dictionary estimates (see also the previous work Arora et al., 2013; Agarwal et al., 2013).

For the undercomplete case (when $r \leq d$), Sun et al. (2017) provide a Riemannian trust region method that can tolerate sparsity $s = \mathcal{O}(d)$, while earlier work by Spielman et al. (2012) provides an algorithm that works in this setting for sparsity $\mathcal{O}(\sqrt{d})$.

Local and global optima of non-convex formulations for the problem have also been extensively studied in (Wu and Yu, 2015; Gribonval et al., 2015; Gribonval and Nielsen, 2003), among others. Apart from alternating minimization, other approaches (without theoretical convergence guarantees) for dictionary learning include K-SVD (Aharon et al., 2006) and MOD (Engan et al., 1999). There is also a nice formulation by Barak et al. (2015), based on the sum-of-squares hierarchy. Recently, Hazan and Ma (2016) provide guarantees for improper dictionary learning, where instead of learning a dictionary, they learn a comparable encoding via convex relaxations. Our work also adds to the recent literature on analyzing alternating minimization algorithms (Jain et al., 2013; Netrapalli et al., 2013, 2014; Hardt, 2014; Balakrishnan et al., 2017).

## 1.2 Contributions

Our main contribution is to present new conditions under which alternating minimization for dictionary learning converges at a linear rate to the optimum. We impose a condition on the matrix infinity norm (largest magnitude entry) of the underlying dictionary. This allows dictionaries with operator norm growing with dimension $(d, r)$. The error rates are measured in the matrix infinity norm, which is sharper than the previous error rates in maximum column-wise error.

Our results hold for a rather arbitrary level of overcompleteness, $r = \mathcal{O}(poly(d))$. We establish convergence results for sparsity level $s = \mathcal{O}(\sqrt{d})$, which is information theoretically optimal for incoherent dictionaries and improves the previously best known results in the overcomplete setting by a logarithmic factor. Our algorithm is simple, involving an $\ell_1$-minimization step followed by a gradient update for the dictionary.

A key step in our proofs is an analysis of a robust sparse estimator—$\{\ell_1, \ell_2, \ell_\infty\}$-MU Selector—under fixed (worst case) corruption in the dictionary. We prove that this estimator is minimax optimal in this setting, which might be of independent interest.

## 1.3 Organization

In Section 2, we present our alternating minimization algorithm and discuss the sparse regression estimator. In Section 3, we list the assumptions under which our algorithm converges and state the main convergence result. Finally, in Section 4, we prove convergence of our algorithm. We defer technical lemmas, analysis of the sparse regression estimator, and minimax analysis to the appendix.

**Notation**

For a vector $v \in \mathbb{R}^d$, $v_i$ denotes the $i^{th}$ component of the vector, $\|v\|_p$ denotes the $\ell_p$ norm, $supp(v)$ denotes the support of a vector $v$, that is, the set of non-zero entries of the vector, $sgn(v)$ denotes

**Algorithm 1:** Alternating Minimization for Dictionary Learning

---

**Input** : Step size $\eta$, samples $\{y^k\}_{k=1}^n$, initial estimate $A^{(0)}$, number of steps $T$, thresholds $\{\tau^{(t)}\}_{t=1}^T$, initial radius $R^{(0)}$ and parameters $\{\gamma^{(t)}\}_{t=1}^T$, $\{\lambda^{(t)}\}_{t=1}^T$ and $\{\nu^{(t)}\}_{t=1}^T$.

1 **for** $t = 1, 2, \ldots, T$ **do**

2      **for** $k = 1, 2, \ldots, n$ **do**

3          $w^{k,(t)} = MUS_{\gamma^{(t)}, \lambda^{(t)}, \nu^{(t)}}(y^k, A^{(t-1)}, R^{(t-1)})$

4          **for** $l = 1, 2, 3 \ldots, r$ **do**

5              $x_l^{k,(t)} = w_l^{k,(t)} \mathbb{I}\left(|w_l^{k,(t)}| > \tau^{(t)}\right), \quad (x^{k,(t)} \text{ is the sparse estimate})$

6          **end**

7      **end**

8      **for** $i = 1, 2, \ldots, d$ **do**

9          **for** $j = 1, 2, \ldots, r$ **do**

10              $A_{ij}^{(t)} = A_{ij}^{(t-1)} - \frac{\eta}{n} \sum_{k=1}^n \left[ \sum_{p=1}^r \left( A_{ip}^{(t-1)} x_p^{k,(t)} x_j^{k,(t)} - y_i^k x_j^{k,(t)} \right) \right]$

11          **end**

12      **end**

13      $R^{(t)} = \frac{7}{8} R^{(t-1)}.$

14 **end**

---

the sign of the vector $v$, that is, a vector with $sgn(v)_i = 1$ if $v_i > 0$, $sgn(v)_i = -1$ if $v_i < 0$ and $sgn(v)_i = 0$ if $v_i = 0$. For a matrix $W$, $W_i$ denotes the $i^{th}$ column, $W_{ij}$ is the element in the $i^{th}$ row and $j^{th}$ column, $\|W\|_{op}$ denotes the operator norm, and $\|W\|_\infty$ denotes the maximum of the magnitudes of the elements of $W$. For a set $J$, we denote its cardinality by $|J|$. Throughout the paper, we use $C$ multiple times to denote global constants that are independent of the problem parameters and dimension. We denote the indicator function by $\mathbb{I}(\cdot)$.

## 2 Algorithm

Given an initial estimate of the dictionary $A^{(0)}$ we alternate between an $\ell_1$ minimization procedure (specifically the $\{\ell_1, \ell_2, \ell_\infty\}$-MU Selector—$MUS_{\gamma, \lambda, \nu}$ in the algorithm—followed by a thresholding step) and a gradient step, under sample $\ell_2$ loss, to update the dictionary. We analyze this algorithm and demand linear convergence at a rate of 7/8; convergence analysis for other rates follows in the same vein with altered constants. Below we state the permitted range for the various parameters in the algorithm above.

1. Step size: We need to set the step size in the range $3r/4s < \eta < r/s$.

2. Threshold: At each step set the threshold at $\tau^{(t)} = 16R^{(t-1)}M(R^{(t-1)}(s+1) + s/\sqrt{d})$.

3. Tuning parameters: We need to pick $\lambda^{(t)}$ and $\nu^{(t)}$ such that the assumption (D5) is satisfied. A choice that is suitable that satisfies this assumption is

$$128s\left(R^{(t-1)}\right)^2 \le \nu^{(t)} \le 3,$$

$$32\left(s^{3/2}\left(R^{(t-1)}\right)^2 + \frac{s^{3/2}R^{(t-1)}}{d^{1/2}}\right)\left(4 + \frac{6}{\sqrt{s}}\right) \le \lambda^{(t)} \le 3.$$

We need to set $\gamma^{(t)}$ as specified by Theorem 16,

$$\gamma^{(t)} = \sqrt{s}\left(R^{(t-1)}\right)^2 + \sqrt{\frac{s}{d}}R^{(t-1)}.$$

### 2.1 Sparse Regression Estimator

Our proof of convergence for Algorithm 1 also goes through with a different choices of robust sparse regression estimators, however, we can establish the tightest guarantees when the $\{\ell_1, \ell_2, \ell_\infty\}$-MU

Selector is used in the sparse regression step. The $\{\ell_1, \ell_2, \ell_\infty\}$-MU Selector (Belloni et al., 2014) was established as a modification of the Dantzig selector to handle uncertainty in the dictionary. There is a beautiful line of work that precedes this that includes (Rosenbaum et al., 2010, 2013; Belloni et al., 2016). There are also modified non-convex LASSO programs that have been studied (Loh and Wainwright, 2011) and Orthogonal Matching Pursuit algorithms under in-variable errors (Chen and Caramanis, 2013). However these estimators require the error in the dictionary to be stochastic and zero mean which makes them less suitable in this setting. Also note that standard $\ell_1$ minimization estimators like the LASSO and Dantzig selector are highly unstable under errors in the dictionary and would lead to much worse guarantees in terms of radius of convergence (as studied in Agarwal et al., 2014). We establish the error guarantees for a robust sparse estimator $\{\ell_1, \ell_2, \ell_\infty\}$-MU Selector under fixed corruption in the dictionary. We also establish that this estimator is minimax optimal when the error in the sparse estimate is measured in infinity norm $\|\hat{\theta} - \theta^*\|_\infty$ and the dictionary is corrupted.

**The $\{\ell_1, \ell_2, \ell_\infty\}$-MU Selector**

Define the estimator $\hat{\theta}$ such that $(\hat{\theta}, \hat{t}, \hat{u}) \in \mathbb{R}^r \times \mathbb{R}_+ \times \mathbb{R}_+$ is the solution to the convex minimization problem

$$\min_{\theta, t, u} \left\{ \|\theta\|_1 + \lambda t + \nu u \, \middle| \, \theta \in \mathbb{R}^r, \left\| \frac{1}{d} A^\top (y - A\theta) \right\|_\infty \leq \gamma t + R^2 u, \|\theta\|_2 \leq t, \|\theta\|_\infty \leq u \right\} \quad (2)$$

where, $\gamma, \lambda$ and $\nu$ are tuning parameters that are chosen appropriately. $R$ is an upper bound on the error in our dictionary measured in matrix infinity norm. Henceforth the first coordinate $(\hat{\theta})$ of this estimator is called $MUS_{\gamma,\lambda,\nu}(y, A, R)$, where the first argument is the sample, the second is the matrix, and the third is the value of the upper bound on the error of the dictionary measured in infinity norm. We will see that under our assumptions we will be able to establish an upper bound on the error on the estimator, $\|\hat{\theta} - \theta^*\|_\infty \leq 16RM \left( R(s+1) + s/\sqrt{d} \right)$, where $|\theta_j^*| \leq M \, \forall j$. We define a threshold at each step $\tau = 16RM(R(s+1) + s/\sqrt{d})$. The thresholded estimate $\tilde{\theta}$ is defined as

$$\tilde{\theta}_i = \hat{\theta}_i \mathbb{I}[|\hat{\theta}_i| > \tau] \qquad \forall i \in \{1, 2, \ldots, r\}. \quad (3)$$

Our assumptions will ensure that we have the guarantee $sgn(\tilde{\theta}) = sgn(\theta^*)$. This will be crucial in our proof of convergence. The analysis of this estimator is presented in Appendix B.

To identify the signs of the sparse covariates correctly using this class of thresholded estimators, we would like in the first step to use an estimator $\hat{\theta}$ that is *optimal*, as this would lead to tighter control over the radius of convergence. This makes the choice of $\{\ell_1, \ell_2, \ell_\infty\}$-MU Selector natural, as we will show it is minimax optimal under certain settings.

**Theorem 1** (informal). *Define the sets of matrices $\mathcal{A} = \{B \in \mathbb{R}^{d \times r} \big| \|B_i\|_2 \leq 1, \, \forall i \in \{1, \ldots, r\}\}$ and $\mathcal{W} = \{P \in \mathbb{R}^{d \times r} \big| \|P\|_\infty \leq R\}$ with $R = \mathcal{O}(1/\sqrt{s})$. Then there exists an $A^* \in \mathcal{A}$ and $W \in \mathcal{W}$ with $A \triangleq A^* + W$ such that*

$$\inf_{\hat{T}} \sup_{\theta^*} \|\hat{T} - \theta^*\|_\infty \geq CRL \left( \sqrt{1 - \frac{\log(s)}{\log(r)}} \right), \quad (4)$$

*where the $\inf_{\hat{T}}$ is over all measurable estimators $\hat{T}$ with input $(A^* \theta^*, A, R)$, and the $\sup$ is over s-sparse vectors $\theta^*$ with 2-norm $L > 0$.*

**Remark 2.** *Note that when $R = \mathcal{O}(1/\sqrt{s})$ and $s = \mathcal{O}(\sqrt{d})$, this lower bound matches the upper bound we have for Theorem 16 (up to logarithmic factors) and hence the $\{\ell_1, \ell_2, \ell_\infty\}$-MU Selector is minimax optimal.*

The proof of this theorem follows by Fano's method and is relegated to Appendix C.

## 2.2 Gradient Update for the dictionary

We note that the update to the dictionary at each step in Algorithm 1 is as follows

$$A_{ij}^{(t)} = A_{ij}^{(t-1)} - \eta \underbrace{\left( \frac{1}{n} \sum_{k=1}^{n} \left[ \sum_{p=1}^{r} \left( A_{ip}^{(t-1)} x_p^{k,(t)} x_j^{k,(t)} - y_i^k x_j^{k,(t)} \right) \right] \right)}_{\triangleq \hat{g}_{ij}^{(t)}},$$

for $i \in \{1, \ldots, d\}$, $j \in \{1, \ldots, r\}$ and $t \in \{1, \ldots, T\}$. If we consider the loss function at time step $t$ built using the vector samples $y^1, \ldots, y^n$ and sparse estimates $x^{1,(t)}, \ldots, x^{n,(t)}$,

$$\mathcal{L}_n(A) = \frac{1}{2n} \sum_{k=1}^{n} \left\| y^k - A x^{k,(t)} \right\|_2^2, \qquad \forall A \in \mathbb{R}^{d \times r},$$

we can identify the update to the dictionary $\hat{g}^{(t)}$ as the gradient of this loss function evaluated at $A^{(t-1)}$,

$$\hat{g}^{(t)} = \left. \frac{\partial \mathcal{L}_n(A)}{\partial A} \right|_{A^{(t-1)}}.$$

## 3 Main Results and Assumptions

In this section we state our convergence result and state the assumptions under which our results are valid.

### 3.1 Assumptions

**Assumptions on $A^*$**

(A1) **Incoherence:** *We assume the the true underlying dictionary is $\mu/\sqrt{d}$-incoherent*

$$|\langle A_i^*, A_j^* \rangle| \leq \frac{\mu}{\sqrt{d}} \quad \forall \, i, j \in \{1, \ldots, r\} \text{ such that, } i \neq j.$$

This is a standard assumption in the sparse regression literature when support recovery is of interest. It was introduced in (Fuchs, 2004; Tropp, 2006) in signal processing and independently in (Zhao and Yu, 2006; Meinshausen and Bühlmann, 2006) in statistics. We can also establish guarantees under the strictly weaker $\ell_\infty$-sensitivity condition (cf. Gautier and Tsybakov, 2011) used in analyzing sparse estimators under in-variable uncertainty in (Belloni et al., 2016; Rosenbaum et al., 2013). The $\{\ell_1, \ell_2, \ell_\infty\}$-MU selector that we use for our sparse recovery step also works with this more general assumption, however for ease of exposition we assume $A^*$ to be $\mu/\sqrt{d}$-incoherent.

(A2) **Normalized Columns**: *We assume that all the columns of $A^*$ are normalized to 1,*

$$\|A_i^*\|_2 = 1 \ \forall \, i \in \{1, \ldots, r\}.$$

Note that the samples $\{y^i\}_{i=1}^{n}$ are invariant when we scale the columns of $A^*$ or under permutations of its columns. Thus we restrict ourselves to dictionaries with normalized columns and label the entire equivalence class of dictionaries with permuted columns and varying signs as $A^*$.

(A3) **Bounded max-norm:** *We assume that $A^*$ is bounded in matrix infinity norm*

$$\|A^*\|_\infty \leq \frac{C_b}{s},$$

*where $C_b = 1/2000M^2$.* This is in contrast with previous work that imposes conditions on the operator norm of $A^*$ (Arora et al., 2015; Agarwal et al., 2014; Arora et al., 2013). Our assumptions help provide guarantees under alternate assumptions and it also allows the operator norm to grow with dimension, whereas earlier work requires $A^*$ to be such that $\|A^*\|_{op} \leq C\left(\sqrt{r/d}\right)$. In general the infinity norm and operator norm balls are hard to

compare. However, one situation where a comparison is possible is if we assume the entries of the dictionary to be drawn iid from a Gaussian distribution $\mathcal{N}(0, \sigma^2)$. Then by standard concentration theorems, for the operator norm condition to be satisfied we would need the variance ($\sigma^2$) of the distribution to scale as $\mathcal{O}(1/d)$ while, for the infinity norm condition to be satisfied we need the variance to be $\tilde{\mathcal{O}}(1/s^2)$. This means that modulo constants the variance can be much larger for the infinity norm condition to be satisfied than for the operator norm condition.

(A4) **Separation:** *We assume that* $\forall i \in \{1, \ldots, r\}$

$$\|A_i^*\|_\infty > \frac{3C_b}{4s}, \text{ and, } \min_{z \in \{-1,1\}} \|A_i^* - zA_j^*\|_\infty \geq \frac{3C_b}{2s} \qquad \forall j \neq i \in \{1, \ldots, r\}.$$

This condition ensures that two dictionaries in the equivalence class with varying signs of columns or permutations are separated in infinity norm. The first condition ensures that for any column $A_i^*$ and $-A_i^*$ are separated $\|A_i^* - (-A_i^*)\|_\infty \geq 3C_b/2s$.

**Assumption on the initial estimate and initialization**

(B1) *We require an initial estimate for the dictionary* $A^{(0)}$ *that is close in infinity norm,*

$$\|A^{(0)} - A^*\|_\infty \leq \frac{C_b}{2s}.$$

This initialization combined with the separation condtion above ensures that the initial estimate $A^{(0)}$ is close to *only one* dictionary in the equivalence class. The algorithm is going to be contractive, hence this will hold true throughout the run of the algorithm.

**Assumptions on** $x^*$

Next we assume a generative model on the $s$-sparse covariates $x^*$. Here are the assumptions we make about the (unknown) distribution of $x^*$.

(C1) **Conditional Independence:** *We assume that distribution of non-zero entries of* $x^*$ *is conditionally independent and identically distributed. That is,* $x_i^* \perp\!\!\!\perp x_j^* | x_i^*, x_j^* \neq 0$.

(C2) **Sparsity Level:***We assume that the level of sparsity $s$ is bounded*

$$2 \leq s \leq \min(2\sqrt{d}, C_b\sqrt{d}, C\sqrt{d}/\mu),$$

*where $C$ is an appropriate global constant such that $A^*$ satisfies assumption (D3), see Remark 15.* For incoherent dictionaries, this upper bound is tight up to constant factors for sparse recovery to be feasible (Donoho and Huo, 2001; Gribonval and Nielsen, 2003).

(C3) **Boundedness:** *Conditioned on the event that $i$ is in the subset of non-zero entries, we have*

$$m \leq |x_i^*| \leq M,$$

*with $m \geq 32R^{(0)}M(R^{(0)}(s+1) + s/\sqrt{d})$ and $M > 1$.* This is needed for the thresholded sparse estimator to correctly predict the sign of the true covariate ($sgn(x) = sgn(x^*)$). We can also relax the boundedness assumption: it suffices for the $x_i^*$ to have sub-Gaussian distributions.

(C4) **Probability of support:** *The probability of $i$ being in the support of $x^*$ is uniform over all $i \in \{1, 2, \ldots, r\}$.* This translates to

$$\mathbb{P}r(x_i^* \neq 0) = \frac{s}{r} \qquad \forall i \in \{1, \ldots, r\},$$

$$\mathbb{P}r(x_i^*, x_j^* \neq 0) = \frac{s(s-1)}{r(r-1)} \qquad \forall i \neq j \in \{1, \ldots, r\}.$$

(C5) **Mean and variance of variables in the support:** *We assume that the non-zero random variables in the support of $x^*$ are centered and are normalized*

$$\mathbb{E}(x_i^* | x_i^* \neq 0) = 0, \qquad \mathbb{E}(x_i^{*2} | x_i^* \neq 0) = 1.$$

We note that these assumptions (A1), (A2) and (C1) - (C5) are similar to those made in (Arora et al., 2015; Agarwal et al., 2014). Agarwal et al. (2014) require the matrices to satisfy the restricted isometry property, which is strictly weaker than $\mu/\sqrt{d}$-incoherence, however they can tolerate a much lower level of sparsity ($d^{1/6}$).

## 3.2 Main Result

**Theorem 3.** *Suppose that true dictionary $A^*$ and the distribution of the $s$-sparse samples $x^*$ satisfy the assumptions stated in Section 3.1 and we are given an estimate $A^{(0)}$ such that $\|A^{(0)} - A^*\|_\infty \leq R^{(0)} \leq C_b/2s$. If we are given $\{n^{(t)}\}_{t=1}^T$ i.i.d. samples in every iteration with $n^{(t)} = \Omega\left(\frac{r}{s(R^{(t-1)})^2}\log(dr/\delta)\right)$ then Algorithm 1 with parameters $(\{\tau^{(t)}\}_{t=1}^T, \{\gamma^{(t)}\}_{t=1}^T, \{\lambda^{(t)}\}_{t=1}^T, \{\nu^{(t)}\}_{t=1}^T, \eta)$ chosen as specified in Section 3.1 after $T$ iterations returns a dictionary $A^{(T)}$ such that,*

$$\|A^{(T)} - A^*\|_\infty \leq \left(\frac{7}{8}\right)^T R^{(0)}, \qquad \text{with probability } 1 - T\delta.$$

## 4 Proof of Convergence

In this section we prove the main convergence result. To prove this we analyze the gradient update to the dictionary at each step. We can decompose this gradient update (which is a random variable) into a first term which is its expected value and a second term which is its deviation from expectation. We will prove a deterministic convergence result by working with the expected value of the gradient and then appeal to standard concentration arguments to control the deviation of the gradient from its expected value with high probability.

By Lemma 8, Algorithm 1 is guaranteed to estimate the sign pattern correctly at every round of the algorithm, $sgn(x) = sgn(x^*)$ (see proof in Appendix A.1). Also note that by assumption (B1), the initial dictionary $A^{(0)}$ is close to *only one* dictionary $A^*$ in the equivalence class.

To un-clutter notation let, $A_{ij}^* = a_{ij}^*, A_{ij}^{(t)} = a_{ij}, A_{ij}^{(t+1)} = a_{ij}'$. The $k^{th}$ coordinate of the $m^{th}$ covariate is written as $x_k^{m*}$. Similarly let $x_k^m$ be the $k^{th}$ coordinate of the estimate of the $m^{th}$ covariate at step $t$. Finally let $R^{(t)} = R$, $n^{(t)} = n$ and $\hat{g}_{ij}$ be the $(i,j)^{th}$ element of the gradient with $n$ ($n^{(t)}$) samples at step $t$. Unwrapping the expression for $\hat{g}_{ij}$ we get,

$$\hat{g}_{ij} = \frac{1}{n}\sum_{m=1}^n\left[\sum_{k=1}^r\left(a_{ik}x_k^m x_j^m\right) - y_i^m x_j^m\right] = \frac{1}{n}\sum_{m=1}^n\left[\sum_{k=1}^r\left(a_{ik}x_k^m - a_{ik}^* x_k^{m*}\right)x_j^m\right]$$

$$= \mathbb{E}\left[\sum_{k=1}^r\left(a_{ik}x_k - a_{ik}^* x_k^*\right)x_j\right]$$

$$+ \left[\frac{1}{n}\sum_{m=1}^n\left[\sum_{k=1}^r\left(a_{ik}x_k^m - a_{ik}^* x_k^{m*}\right)x_j^m\right] - \mathbb{E}\left[\sum_{k=1}^r\left(a_{ik}x_k - a_{ik}^* x_k^*\right)x_j\right]\right]$$

$$= g_{ij} + \underbrace{\hat{g}_{ij} - g_{ij}}_{\triangleq \epsilon_n},$$

where $g_{ij}$ denotes $(i,j)^{th}$ element of the expected value (infinite samples) of the gradient. The second term $\epsilon_n$ is the deviation of the gradient from its expected value. By Theorem 10 we can control the deviation of the sample gradient from its mean via an application of McDiarmid's inequality. With this notation in place we are now ready to prove Theorem 3.

**Proof** [Proof of Theorem 3] First we analyze the structure of the expected value of the gradient.

*Step 1*: Unwrapping the expected value of the gradient we find it decomposes into three terms

$$g_{ij} = \mathbb{E}\left(a_{ij}x_j^2 - a_{ij}^* x_j^* x_j\right) + \mathbb{E}\left[\sum_{k\neq j}a_{ik}x_k x_j - a_{ik}^* x_k^* x_j\right]$$

$$= \underbrace{(a_{ij} - a_{ij}^*)\frac{s}{r}\mathbb{E}\left[x_j^2|x_j^* \neq 0\right]}_{\triangleq g_{ij}^c} + \underbrace{a_{ij}^*\frac{s}{r}\mathbb{E}\left[(x_j - x_j^*)x_j|x_j^* \neq 0\right]}_{\triangleq \Xi_1} + \underbrace{\mathbb{E}\left[\sum_{k\neq j}a_{ik}x_k x_j - a_{ik}^* x_k^* x_j\right]}_{\triangleq \Xi_2}.$$

The first term $g_{ij}^c$ points in the *correct direction* and will be useful in converging to the right answer. The other terms could be in a bad direction and we will control their magnitude with Lemma 5 such that $|\Xi_1| + |\Xi_2| \leq \frac{s}{3r}R$. The proof of Lemma 5 is the main technical challenge in the convergence analysis to control the error in the gradient. Its proof is deferred to the appendix.

*Step 2*: Given this bound, we analyze the gradient update,

$$a'_{ij} = a_{ij} - \eta\hat{g}_{ij} = a_{ij} - \eta(g_{ij} + \epsilon_n) = a_{ij} - \eta\left[g_{ij}^c + (\Xi_1 + \Xi_2) + \epsilon_n\right].$$

So if we look at the distance to the optimum $a_{ij}^*$ we have the relation,

$$a'_{ij} - a_{ij}^* = a_{ij} - a_{ij}^* - \eta(a_{ij} - a_{ij}^*)\frac{s}{r}\mathbb{E}\left[x_j^2|x_j^* \neq 0\right] - \eta\left\{(\Xi_1 + \Xi_2) + \epsilon_n\right\}.$$

Taking absolute values, we get

$$|a'_{ij} - a_{ij}^*| \overset{(i)}{\leq} \left(1 - \eta\frac{s}{r}\mathbb{E}\left[x_j^2|x_j^* \neq 0\right]\right)|a_{ij} - a_{ij}^*| + \eta\left\{|\Xi_1| + |\Xi_2| + |\epsilon_n|\right\}$$

$$\overset{(ii)}{\leq} \left(1 - \eta\frac{s}{r}\mathbb{E}\left[x_j^2|x_j^* \neq 0\right]\right)|a_{ij} - a_{ij}^*| + \eta\left(\frac{s}{3r}R\right) + \eta|\epsilon_n|$$

$$\leq \left(1 - \eta\frac{s}{r}\left\{\mathbb{E}\left[x_j^2|x_j^* \neq 0\right] - \frac{1}{3}\right\}\right)R + \eta|\epsilon_n|,$$

provided the first term is at non-negative. Here, $(i)$ follows by triangle inequality and $(ii)$ is by Lemma 5. Next we give an upper and lower bound on $\mathbb{E}\left[x_j^2|x_j^* \neq 0\right]$. We would expect that as $R$ gets smaller this variance term approaches $\mathbb{E}\left[x_j^{*2}|x_j^* \neq 0\right] = 1$. By invoking Lemma 6 we can bound this term to be $\frac{2}{3} \leq \mathbb{E}\left[x_j^2|x_j^* \neq 0\right] \leq \frac{4}{3}$. We want the first term to contract at a rate $3/4$; it suffices to have

$$0 \overset{(i)}{\leq} \left(1 - \eta\frac{s}{r}\left\{\mathbb{E}\left[x_j^2|x_j^* \neq 0\right] - \frac{1}{3}\right\}\right) \overset{(ii)}{\leq} \frac{3}{4}.$$

Coupled with Lemma 6, Inequality $(i)$ follows from $\eta \leq \frac{r}{s}$ while $(ii)$ follows from $\eta \geq \frac{3r}{4s}$. We also have by Theorem 10 that $\eta|\epsilon_n| \leq R/8$ with probability $1 - \delta$. So if we unroll the bound for $t$ steps we have,

$$|a_{ij}^{(t)} - a_{ij}^*| \leq \frac{3}{4}R^{(t-1)} + \eta|\epsilon_n| \leq \frac{3}{4}R^{(t-1)} + \frac{1}{8}R^{(t-1)} = \frac{7}{8}R^{(t-1)} \leq \left(\frac{7}{8}\right)^t R^{(0)}.$$

We also have $\eta|\epsilon_n| \leq R/8 \leq R^{(0)}/8$ with probability at least $1 - \delta$ in each iteration, for all $t \in \{1, \ldots, T\}$; thus by taking a union bound over the iterations we are guaranteed to remain in our initial ball of radius $R^{(0)}$ with high probability, completing the proof. ∎

# 5   Conclusion

An interesting question would be to further explore and analyze the range of algorithms for which alternating minimization works and identifying the conditions under which they provably converge. Going beyond sparsity $\sqrt{d}$ still remains challenging, and as noted in previous work alternating minimization also appears to break down experimentally and new algorithms are required in this regime. Also all theoretical work on analyzing alternating minimization for dictionary learning seems to rely on identifying the signs of the samples ($x^*$) correctly at every step. It would be an interesting theoretical question to analyze if this is a necessary condition or if an alternate proof strategy and consequently a bigger radius of convergence are possible. Lastly, it is not known what the optimal sample complexity for this problem is and lower bounds there could be useful in designing more sample efficient algorithms.

**Acknowledgments**

We gratefully acknowledge the support of the NSF through grant IIS-1619362, and of the Australian Research Council through an Australian Laureate Fellowship (FL110100281) and through the ARC

Centre of Excellence for Mathematical and Statistical Frontiers. Thanks also to the Simons Institute for the Theory of Computing Spring 2017 Program on Foundations of Machine Learning.

The authors would like to thank Sahand Negahban for pointing out an error in the $\mu$-incoherence assumption in an earlier version. The authors would like to thank Shivam Garg for pointing us to an error in the claim about random initialization in a previous version of this paper.

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
