[Supplementary Material · Niladri_supplementary_material.pdf]

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

# A  Additional details for the proof of convergence

For Appendix A.1 and A.2, we borrow the notation from Section 4. In Appendix A.1 we prove Lemma 4 that controls an error term which will be useful in establishing Lemma 5 that bounds the error terms in the gradient, $\Xi_1$ and $\Xi_2$. Corollary 9 establishes the error bound for the sparse estimate while Lemma 8 establishes that the sparse estimate after the thresholding step has the correct sign. In Appendix A.2, we establish finite sample guarantees.

## A.1  Proof of Auxillary Lemmas

Before we prove Lemma 5, which controls the terms in the gradient, we prove Lemma 4, which will be vital in controlling the cross-term in the gradient.

**Lemma 4.** *Let the assumptions stated in Section 3.1 hold. Then at each iteration step we have the guarantee that*

$$\left| \max_{k:k\neq j} \left\{ \mathbb{E}\left[ a_{ik}x_k x_j - a_{ik}^* x_k^* x_j | x_k^* \neq 0, x_j^* \neq 0 \right] \right\} \right| \leq \frac{R}{6(s-1)}.$$

**Proof** Let us define

$$\Gamma \triangleq \max_{k:k\neq j} \left\{ \mathbb{E}\left[ a_{ik}x_k x_j - a_{ik}^* x_k^* x_j | x_k^* \neq 0, x_j^* \neq 0 \right] \right\},$$

and let us define the event $\mathcal{E}_{jk} \triangleq \{x_j^* \neq 0, x_k^* \neq 0\}$. Expanding $\Gamma$,

$$\Gamma = \max_{k:k\neq j} \left\{ \mathbb{E}\left[ a_{ik}(x_k - x_k^* + x_k^*)(x_j - x_j^* + x_j^*) - a_{ik}^* x_k^*(x_j - x_j^* + x_j^*) | \mathcal{E}_{jk} \right] \right\}$$

$$= \max_{k:k\neq j} \left\{ \underbrace{(a_{ik} - a_{ik}^*)\mathbb{E}\left[ x_k^*(x_j - x_j^*) | \mathcal{E}_{jk} \right]}_{\triangleq n_1} + \underbrace{a_{ik}\mathbb{E}\left[ (x_k - x_k^*)x_j^* | \mathcal{E}_{jk} \right]}_{\triangleq n_2} \right.$$

$$\left. + \underbrace{a_{ik}\mathbb{E}\left[ (x_k - x_k^*)(x_j - x_j^*) | \mathcal{E}_{jk} \right]}_{\triangleq n_3} + \underbrace{(a_{ik} - a_{ik}^*)\mathbb{E}\left[ x_k^* x_j^* | \mathcal{E}_{jk} \right]}_{\triangleq n_4} \right\}.$$

Given that the non-zero entries of $x^*$ are independent and mean zero we have $n_4 = 0$. Next we see $n_1, n_2$ and $n_3$ are bounded above as

$$n_1 \leq |a_{ik} - a_{ik}^*| M \|x - x^*\|_\infty \leq RM\|x - x^*\|_\infty$$
$$n_2 \leq |a_{ik}| M \|x - x^*\|_\infty \leq (|a_{ik}^*| + R)M\|x - x^*\|_\infty$$
$$n_3 \leq |a_{ik}| \|x - x^*\|_\infty^2 \leq (R + |a_{ik}^*|)\|x - x^*\|_\infty^2,$$

these follow as $|x_j^*| \leq M$, $|x_j - x_j^*| \leq \|x - x^*\|_\infty$ and $|a_{ik} - a_{ik}^*| \leq R$. The goal now is to show that $n_1 \leq R/30(s-1)$, $n_2 \leq R/15(s-1)$ and $n_3 \leq R/15(s-1)$. Let us unwrap the first term of $n_1$

$$n_1 \leq RM\|x - x^*\|_\infty \overset{(i)}{\leq} \frac{R}{30(s-1)}\left[ 30(s-1)M \cdot 16RM\left( R(s+1) + \frac{s}{\sqrt{d}} \right) \right]$$

$$\overset{(ii)}{\leq} \frac{R}{30(s-1)}\left[ 30(s-1)M \cdot \frac{8C_b M}{s}\left( \frac{C_b(s+1)}{2s} + 2 \right) \right]$$

$$= \frac{R}{30(s-1)}\left[ 240M^2 \underbrace{\left( \frac{s-1}{s} \right)}_{\leq 1} C_b \left( C_b \underbrace{\left( \frac{(s+1)}{2s} \right)}_{\leq 3/4} + 2 \right) \right]$$

$$\leq \frac{R}{30(s-1)}\underbrace{\left[ 240M^2 C_b \left( \frac{3}{4}C_b + 2 \right) \right]}_{\triangleq \xi_1}, \tag{5}$$

where $(i)$ follows by invoking Corollary 9 and $(ii)$ follows as $s \le 2\sqrt{d}$ and $R \le C_b/2s$. Our choice $C_b = 1/2000M^2$ ensures that $\xi_1 \le 1$. The second term in the upper bound on $n_2$ can be bounded by the same technique as we used to bound $n_1$, giving $RM\|x - x^*\|_\infty \le R/30(s-1)$. For the first term in $n_2$, we have

$$|a_{ik}^*|M\|x - x^*\|_\infty \le \frac{R}{30(s-1)}\left[480\frac{(s-1)}{s}M^2 C_b\left(R(s+1) + \frac{s}{\sqrt{d}}\right)\right]$$

$$\le \frac{R}{30(s-1)}\underbrace{\left[480M^2 C_b\left(C_b\frac{(s+1)}{2s} + 2\right)\right]}_{\triangleq \xi_2}, \tag{6}$$

where these inequalities follow by invoking Corollary 9 and by the upper bounds on $|a_{ik}^*|$ and $R$. Again our choice $C_b = 1/2000M^2$ ensures that $\xi_2 \le 1$ which leaves us with the upper bound on $n_2 \le \frac{R}{15(s-1)}$. Finally to bound $n_3$ we observe that the first term is bounded as follows,

$$R\|x - x^*\|_\infty^2 \le \frac{R}{30(s-1)}\left[30(s-1)\cdot 16^2 R^2 M^2\left(R(s+1) + \frac{s}{\sqrt{d}}\right)^2\right]$$

$$\le \frac{R}{30(s-1)}\left[\sqrt{30(s-1)}\cdot 16\frac{C_b}{2s}M\left(C_b\frac{(s+1)}{2s} + 2\right)\right]^2$$

$$\le \frac{R}{30(s-1)},$$

where the last inequality is due to the fact that $\xi_1 \le 1$. We have $|a_{ik}^*| \le C_b/s$ and similar arguments as above can be used to show that the second term in $n_3$ is also bounded above by $\frac{R}{30(s-1)}$. Having controlled $n_1, n_2$ and $n_3$ at the appropriate levels completes the proof and yields the desired bound on $\Gamma$. ∎

**Lemma 5.** *Let the assumptions stated in Section 3.1 hold. Then at each iteration step we can bound the error terms in the gradient as*

$$|\Xi_1| = \left|a_{ij}^*\frac{s}{r}\mathbb{E}\left[(x_j - x_j^*)x_j|x_j^* \ne 0\right]\right| \le \frac{s}{6r}R$$

$$|\Xi_2| = \left|\mathbb{E}\left[\sum_{k \ne j}a_{ik}x_k x_j - a_{ik}^*x_k^*x_j\right]\right| \le \frac{s}{6r}R.$$

**Proof** *Part 1*-We first prove the bound on $\Xi_1$. We start by unpacking $\Xi_1$

$$|\Xi_1| = \left|\frac{s}{r}a_{ij}^*\mathbb{E}\left[(x_j - x_j^*)x_j|x_j^* \ne 0\right]\right|$$

$$\le \frac{s}{r}|a_{ij}^*| \cdot \left|\mathbb{E}\left[(x_j - x_j^*)(x_j^* + x_j - x_j^*)|x_j^* \ne 0\right]\right|$$

$$\overset{(i)}{\le} \frac{s}{r}|a_{ij}^*|M \cdot \mathbb{E}\left[\|x - x^*\|_\infty|x_j^* \ne 0\right] + \frac{s}{r}|a_{ij}^*|\mathbb{E}\left[\|x - x^*\|_\infty^2|x_j^* \ne 0\right]$$

$$\overset{(ii)}{\le} \frac{s}{r}|a_{ij}^*|M \cdot \left(16RM\left((s+1)R + \frac{s}{\sqrt{d}}\right)\right) + \frac{s}{r}|a_{ij}^*|\left(16RM\left((s+1)R + \frac{s}{\sqrt{d}}\right)\right)^2$$

$$= \frac{s}{6r}R\left\{96|a_{ij}^*|M^2\left(R(s+1) + \frac{s}{\sqrt{d}}\right) + 6|a_{ij}^*|R\left(16M\left(R(s+1) + \frac{s}{\sqrt{d}}\right)\right)^2\right\} \tag{7}$$

$$\le \frac{s}{6r}R,$$

where $(i)$ follows by triangle inequality and $|x_j^*| \le M$ and, $(ii)$ follows by Corollary 9. It can be shown that in (7) the term in the curly braces is $\le 1$ by arguments similar to those used in Lemma 4 (because $R \le 1/4000M^2s$, $s \le 2\sqrt{d}$ and $|a_{ij}^*| \le 1/2000M^2s$), thus establishing the desired bound on $|\Xi_1|$.

*Part 2-* Expanding $\Xi_2$ we find

$$
\begin{aligned}
|\Xi_2| &= \left| \mathbb{E}\left[ \sum_{k \neq j} a_{ik}x_k x_j - a_{ik}^* x_k^* x_j^* \right] \right| \\
&\stackrel{(i)}{=} \frac{s(s-1)}{r(r-1)} \left| \mathbb{E}\left[ \sum_{k \neq j} a_{ik}x_k x_j - a_{ik}^* x_k^* x_j | x_k^* \neq 0, x_j^* \neq 0 \right] \right| \\
&\leq \frac{s(s-1)}{r(r-1)} \cdot (r-1) \left| \max_{k \neq j} \left\{ \mathbb{E}\left[ a_{ik}x_k x_j - a_{ik}^* x_k^* x_j | x_k^* \neq 0, x_j^* \neq 0 \right] \right\} \right| \\
&= \frac{s}{6r} R \left( \frac{6(s-1)}{R} \left| \max_{k \neq j} \left\{ \mathbb{E}\left[ a_{ik}x_k x_j - a_{ik}^* x_k^* x_j | x_k^* \neq 0, x_j^* \neq 0 \right] \right\} \right| \right) \\
&\stackrel{(ii)}{\leq} \frac{s}{6r} R,
\end{aligned}
$$

where $(i)$ follows from assumption (C4) and $(ii)$ follows by invoking Lemma 4.  ∎

**Lemma 6.** *Let the assumptions stated in Section 3.1 hold. Then at each iteration step we can bound the variance of the estimate,*

$$
\frac{2}{3} \leq \mathbb{E}\left[ x_j^2 | x_j^* \neq 0 \right] \leq \frac{4}{3}.
$$

**Proof** Consider the expectation of the random variable $x_j^2 - x_j^{*2} | x_j^* \neq 0$. We have

$$
\begin{aligned}
x_j^2 - x_j^{*2} &\leq |x_j + x_j^*| \|x - x^*\|_\infty \\
&= |2x_j^* + x_j - x_j^*| \|x - x^*\|_\infty \leq 2|x_j^*| \|x - x^*\|_\infty + \|x - x^*\|_\infty^2 \\
&\leq \underbrace{2M\|x - x^*\|_\infty + \|x - x^*\|_\infty^2}_{\triangleq \xi_3}.
\end{aligned}
$$

Note that $\xi_3 \leq \frac{1}{3}$, if $\|x - x^*\|_\infty \leq \frac{1}{3}\left(\sqrt{3}\sqrt{3M^2 + 1} - 3M\right)$. We also have an upper bound on $\|x - x^*\|_\infty$ by Corollary 9

$$
\begin{aligned}
\|x - x^*\|_\infty &\leq 16RM\left( R(s+1) + \frac{s}{\sqrt{d}} \right) \leq \underbrace{\frac{8}{s}}_{\leq 4} C_b M \left( C_b \underbrace{\frac{s+1}{2s}}_{\leq 3/4} + \underbrace{\frac{s}{\sqrt{d}}}_{\leq 2} \right) \\
&\leq 4C_b M \left( \frac{3}{4} C_b + 2 \right).
\end{aligned}
$$

Our choice $C_b = 1/2000M^2$ with $M > 1$ guarantees that

$$
4C_b M \left( \frac{3}{4} C_b + 2 \right) \leq \frac{1}{3}\left( \sqrt{3}\sqrt{3M^2 + 1} - 3M \right), \tag{8}
$$

this yields the claimed bound.  ∎

The next corollary establishes an infinity norm bound on the error in the sparse estimate under the assumptions made in Section 3.1 and choice of parameters in Section 2.

**Corollary 7.** *Under the assumptions specified in Section 3.1 and choice of parameters for Algorithm 1 in Section 2 we have the bound for all $t \in \{1, \ldots, T\}$ and $k \in \{1, \ldots, n\}$,*

$$
\|w^{k,(t)} - x^{k*}\|_\infty \leq 16R^{(t-1)}M\left( R^{(t-1)}(s+1) + \frac{s}{\sqrt{d}} \right),
$$

*where $w^{k,(t)}$ is as defined in Algorithm 1.*

**Proof** We have $\|x^{k*}\|_2 \le \sqrt{s}M$, $\|x^{k*}\|_\infty \le M$ thus plugging this into Theorem 16 gives us the desired result. ∎

The next theorem guarantees that at each round of the algorithm, under the assumptions stated in Section 3.1, we correctly predict the sign pattern.

**Lemma 8.** *Under the assumptions (A1)-(A6),(B1) and (C1)-(C5) stated in Section 3.1 with*

$$32R^{(0)}M\left(R^{(0)}(s+1) + \frac{s}{\sqrt{d}}\right) < m, \tag{9}$$

*and under the choice of the parameters $\eta, R^{(t)}, \tau^{(t)}, \gamma^{(t)}, \lambda^{(t)}$ and $\nu^{(t)}$ specified in Section 2 for all $t \in \{1, 2, \ldots, T\}$ we have the guarantee that Algorithm 1 returns a sparse estimate $\{x^{k,(t)}\}_{k=1}^n$ such that,*

$$sgn(x^{k,(t)}) = sgn(x^{k*}), \qquad\qquad \forall k \in \{1, 2, \ldots, n\}.$$

**Proof** Under the assumptions stated we can invoke Corollary 7 to get,

$$\|w^{k,(t)} - x^{k*}\|_\infty \le 16R^{(t-1)}M\left(R^{(t-1)}(s+1) + \frac{s}{\sqrt{d}}\right) \quad \forall k \in \{1, \ldots, n\}, t \in \{1, \ldots, T\}, \tag{10}$$

where $w^{k,(t)}$ is defined as in Algorithm 1. Note that the thresholds are defined by the schedule,

$$\tau^{(t)} = 16R^{(t-1)}M\left(R^{(t-1)}(s+1) + \frac{s}{\sqrt{d}}\right).$$

By definition $x^{k,(t)}$ is the coordinate-wise thresholded estimate,

$$x_l^{k,(t)} = w_l^{k,(t)}\mathbb{I}\left(|w_l^{k,(t)}| > \tau^{(t)}\right) \qquad\qquad \forall l \in \{1, 2, \ldots, r\}.$$

We know that for all $t > 1$ we have $R^{(t)} < R^{(0)}$. So by the infinity norm bound in the above display (10) and, by the assumptions on the distribution of $x^*$, we have that

$$sgn\left(x^{k,(t)}\right) = sgn\left(x^{k*}\right) \qquad\qquad \forall k \in \{1, 2, \ldots, n\}.$$

This follows as the thresholding step only zeros out the non-zero elements in $x^{k,(t)}$ that are not in $supp(x^{k*})$. ∎

**Corollary 9.** *Under the assumptions specified in Section 3.1 and choice of parameters for Algorithm 1 in Section 2 we have the bound for all $t \in \{1, \ldots, T\}$ and $k \in \{1, \ldots, n\}$,*

$$\|x^{k,(t)} - x^{k*}\|_\infty \le 16R^{(t-1)}M\left(R^{(t-1)}(s+1) + \frac{s}{\sqrt{d}}\right),$$

*where $x^{k,(t)}$ is as defined in Algorithm 1.*

**Proof** Note that by Lemma 8 we have that $sgn(x^{k,(t)}) = sgn(x^{k*})$. Thus for any $l \in \{1, \ldots, r\}$ if $l \notin supp(x^{k*})$ then the choice of threshold of $\tau^{(t)} = 16R^{(t-1)}M\left(R^{(t-1)}(s+1) + \frac{s}{\sqrt{d}}\right)$ implies that,

$$|x_l^{k,(t)} - x_l^{k*}| = |x_l^{k,(t)}| \le 16R^{(t-1)}M\left(R^{(t-1)}(s+1) + \frac{s}{\sqrt{d}}\right).$$

While for $l \in supp(x^{k*})$ Corollary 7 implies

$$|x_l^{k,(t)} - x_l^{k*}| \le 16R^{(t-1)}M\left(R^{(t-1)}(s+1) + \frac{s}{\sqrt{d}}\right).$$

This completes the proof. ∎

## A.2 Finite Sample Guarantees

In this section, we establish finite sample guarantees and state convergence results used in the proof of convergence of our algorithm.

**Theorem 10.** *Let $\epsilon_n \leq \frac{R}{8\eta}$, where $\frac{3r}{4s} \leq \eta \leq \frac{r}{s}$ is the step-size used at each gradient step. If we are given $n$ i.i.d. samples at each round where $n = \Omega(\frac{r}{sR^2}\log(dr/\delta))$, then we have the guarantee that*

$$\max_{i \in \{1,\dots,r\}, j \in \{1,\dots,d\}} \{|\hat{g}_{ij} - g_{ij}|\} \leq \epsilon_n,$$

*with probability $1 - \delta$.*

**Proof** We define the set $W = \{m : j \in supp(x^{m*})\}$ and then we have that

$$\hat{g}_{ij} = \frac{|W|}{n} \cdot \underbrace{\frac{1}{|W|} \sum_{m \in W} \left( \sum_k a_{ik} x_k^m - a_{ik}^* x_k^{m*} \right) x_j^m}_{\triangleq \hat{g}_{ij}^W}.$$

Let $x^{l*}$ be a sample such that $l \in W$. We will bound the term $\Lambda = |\left(\sum_k a_{ik} x_k^l - a_{ik}^* x_k^{l*}\right) x_j^l|$ and later invoke McDiarmid's inequality. To ease notation we drop the superscript $l$. Expanding $\Lambda$ we get

$$\Lambda = \left| \sum_{k=1}^r (a_{ik} - a_{ik}^*)(x_k - x_k^*)(x_j - x_j^*) + (a_{ik} - a_{ik}^*)(x_k - x_k^*)x_j^* \right.$$

$$\left. + (a_{ik} - a_{ik}^*)x_k^*(x_j - x_j^*) + (a_{ik} - a_{ik}^*)x_k^* x_j^* - a_{ik}^* x_k^*(x_j - x_j^*) - a_{ik}^* x_k^* x_j^* \right|$$

Recall that by Lemma 8 we have that $sgn(x^l) = sgn(x^{l*})$, and $x^{l*}$ is $s$-sparse thus only $s$ terms in the above sum are non-zero. We repeatedly use the bounds,

1. $|a_{ik}^*| \leq \frac{C_b}{s}$.
2. $|a_{ik} - a_{ik}^*| \leq R \leq R^{(0)} \leq \frac{C_b}{2s}$.
3. $\|x - x^*\|_\infty \leq 16RM\left(R(s+1) + \frac{s}{\sqrt{d}}\right)$.
4. $2 \leq s \leq 2\sqrt{d}$.

Using these we can upper bound $\Lambda$ by

$$\Lambda \quad \leq \quad \frac{3C_b M^2}{4} \quad + \quad \frac{10C_b^2 M^2}{s}\left(\frac{C_b(s+1)}{2s} + 2\right) \quad + \quad \frac{C_b}{2}\left(\frac{8C_b M}{s}\left(\frac{C_b(s+1)}{2s} + 2\right)\right)^2.$$

By our choice of $C_b = 1/2000M^2$, where $M > 1$ we have that

$$\Lambda \leq B,$$

for an appropriate global constant $B$ (independent of $s$ and $M$).

By simple concentration arguments we can get that $|W|/n$ is close to $s/r$. Conditioned on a value of $|W|$ by invoking McDiarmid's inequality (Theorem 11), we have that $|\hat{g}_{ij}^W - \mathbb{E}\left[\hat{g}_{ij}|j \in supp(x^*)\right]| \leq \epsilon_{W,n}$ with probability $1 - 2e^{-2|W|\epsilon_{W,n}^2/B^2}$. We demand

$$\epsilon_{W,n} = \frac{C \cdot r \cdot R}{8s\eta}, \tag{11}$$

with probability $1 - c\delta/dr$ for every $(i,j)$, where $c$ and $C$ are appropriate constants such that $|\hat{g}_{ij} - g_{ij}| \leq R/8\eta$ with probability at least $1 - \delta/dr$. Thus we need $|W| = \Omega((\frac{s\eta}{rR})^2 \log(dr/\delta))$. As $\eta$ is proportional to $r/s$, this implies that for (11) to hold, we need that $|W| = \Omega(1/R^2 \log(dr/\delta))$.

As stated above we have that $|W|/n$ is close to $s/r$ so if $|W| = \Omega(1/R^2 \log(dr/\delta))$ it suffices to have $n = \Omega(\frac{r}{sR^2}\log(dr/\delta))$. We finish the proof by a union bound over all entries of the matrix. ∎

### A.3 Concentration Theorems

We recall McDiarmid's inequality (McDiarmid, 1989).

**Theorem 11.** *Let $X_1, \ldots, X_m$ be independent random variables all taking values in the set $\mathcal{X}$. Further, let $f : \mathcal{X}^m \mapsto \mathbb{R}$ be a function of $X_1, X_2, \ldots, X_m$ that satisfies $\forall i, \forall x_1, \ldots, x_m, x_i' \in \mathcal{X}$,*

$$|f(x_1, \ldots, x_i, \ldots, x_m) - f(x_1, \ldots, x_i', \ldots, x_m)| \leq c_i.$$

*Then for all $\epsilon > 0$,*

$$\mathbb{P}(f - \mathbb{E}[f] \geq \epsilon) \leq \exp\left(\frac{-2\epsilon^2}{\sum_{i=1}^m c_i^2}\right).$$

Next we present a concentration theorem for a sum of the squares of $d$ independent Gaussian random variables each with variance $\sigma^2$ ($\chi^2$-concentration theorem).

**Theorem 12** (Gaussian concentration inequality, see Theorem 5.6 in (Boucheron et al., 2013))**.** *Let $X = (X_1, \ldots, X_n)$ be a vector of $n$ independent standard normal random variables. Let $f : \mathbb{R}^n \mapsto \mathbb{R}$ denote an $L$-Lipschitz function with respect to Euclidean distance. Then, for all $t > 0$,*

$$\mathbb{P}(f(X) - \mathbb{E}(f(X)) \geq t) \leq e^{-t^2/(2L^2)}.$$

**Lemma 13.** *If $\{Z_k\}_{k=1}^d \sim \mathcal{N}(0,1)$ are i.i.d. standard normal variables, then $Y \triangleq \sigma^2 \sum_{k=1}^d Z_k^2$ is a scaled chi-squared variate with $d$ degrees of freedom. Define $V \triangleq \sqrt{Y}$, then for all $\delta > 0$ we have*

$$\mathbb{P}\left[V \geq \sigma\sqrt{d} + \delta\right] \leq \exp\left(-\frac{\delta^2}{2\sigma^2}\right).$$

**Proof** Note that by definition $V(Z_1, \ldots, Z_d)$ is a $\sigma$-Lipschitz function of $d$ standard normal variables. By Jensen's inequality we have,

$$\mathbb{E}[V] \leq \sqrt{\mathbb{E}[V^2]} = \sigma\sqrt{d}.$$

Thus by applying Theorem 12 to $V$ we have the claimed bound. ∎

## B  Analysis of Robust Sparse Estimator

Analysis of the $\{\ell_1, \ell_2, \ell_\infty\}$-MU Selector (2) is presented in (Belloni et al., 2014), which we adapt here to present guarantees for deterministic (worst case) perturbations to the dictionary. The analysis in (Belloni et al., 2014) is in a setting where the error in the $A$ is random with zero mean. Here, we consider the error to be deterministic (worst case). Let us start by introducing some notation and important definitions.

### B.1  Notation and Definitions

Let $J \subset \{1, \ldots, r\}$ be a set of integers. For a vector $\theta = (\theta_1, \ldots, \theta_r) \in \mathbb{R}^r$ we denote by $\theta_J$ the vector in $\mathbb{R}^r$ whose $j^{th}$ component satisfies $(\theta_J)_j = \theta_j$ if $j \in J$, and $(\theta_J)_j = 0$ otherwise. Let $diag(\cdot)$ be the matrix formed by just the diagonal entries and zeroing out the off diagonal terms. Also let $\Delta \triangleq \hat{\theta} - \theta^*$ and $W \triangleq A - A^*$, where $\theta^*$ is the true parameter and $A^*$ is the true dictionary without error. Define the cone,

$$\mathcal{C}_J(u) \triangleq \{\Delta \in \mathbb{R}^r : \|\Delta_{J^c}\|_1 \leq u\|\Delta_J\|_1\},$$

where $J$ is a subset of $\{1, \ldots, r\}$. For $q \in [1, \infty]$ and an integer $s \in [1, r]$, the $\ell_q$-sensitivity (see for example Gautier and Tsybakov (2011); Rosenbaum et al. (2013); Belloni et al. (2014, 2016)) is defined as

$$\kappa_q(s, u) \triangleq \min_{J:|J|\leq s} \left(\min_{\Delta \in \mathcal{C}_J(u):\|\Delta\|_q=1} \frac{1}{d}\|A^{*\top} A^* \Delta\|_\infty\right).$$

The $\ell_q$-sensitivity is routinely used to study convergence of estimators under sparsity constraints. If we have $\kappa_q(s, u) \geq cs^{-1/q}$ for some constant $c > 0$, this leads to optimal bounds for the errors. It has also been shown to be a strict generalization of the restricted eigenvalue property and of the mutual incoherence condition. Relations between these conditions are provided by Lemma 6 of Belloni et al. (2016). We restate that lemma here.

**Lemma 14** (Restated from Belloni et al. (2016)). *Let $u > 0$. For any $\alpha \in (0, 1)$ there exists a $c > 0$ such that for $1 \leq s \leq r$ and $1 \leq d \leq r$ with $\mu/\sqrt{d} \leq 1/(cs)$ then*

$$\kappa_\infty(s, u) \geq \alpha.$$

*Furthermore, for any $1 \leq q \leq \infty$,*

$$\kappa_q(s, u) \geq \left(\frac{1}{2s}\right)^{1/q} \kappa_\infty(s, u).$$

Next we highlight the assumptions under which we can establish guarantees for this estimator.

## B.2 Assumptions

We make the following assumptions in the analysis of $\{\ell_1, \ell_2, \ell_\infty\}$-MU Selector.

(D1) We assume that the true dictionary $A^*$ is deterministic. We also assume that $A$ is deterministic.

(D2) We assume that the columns of $A^*$ are normalized, that is, $\|A_i^*\|_2 = 1 \ \forall i \in \{1, 2, \ldots, r\}$.

(D3) For the matrix $A^*$ we assume the $\ell_\infty$-sensitivity is bounded below

$$\kappa_\infty(s, 1 + \lambda + \nu) \geq 1/4.$$

(D4) We demand that $\|W\|_\infty \leq R$.

(D5) Finally, the tuning parameters $\lambda$ and $\nu$ are chosen such that

$$8s \underbrace{\left(\frac{\left(\sqrt{s}R^2 + \sqrt{\frac{s}{d}}R\right)\left(1 + \nu + \frac{2\lambda}{\sqrt{s}}\right)}{\lambda} + \frac{R^2(1+\lambda)}{\nu}\right)}_{\triangleq \zeta} \leq \frac{1}{2}.$$

**Remark 15.** *If the dictionary $A^*$ is $\mu/\sqrt{d}$-incoherent and if the sparsity level $s \leq C\sqrt{d}/\mu$ for an appropriate global constant $C$ then by Lemma 14 Assumption (D3) holds for $A^*$.*

**Theorem 16** (Adapted from Belloni et al. (2014)). *Let assumptions (D1) - (D5) hold. Assume that the true parameter $\theta^*$ is $s$-sparse and belongs to $\Theta$. Let $0 < \lambda, \nu < \infty$, $\gamma = \sqrt{s}R^2 + \sqrt{\frac{s}{d}}R$, and let $\hat\theta$ be the $\{\ell_1, \ell_2, \ell_\infty\}$-MU Selector. Then*

$$\|\hat\theta - \theta^*\|_\infty \leq 16(\gamma\|\theta^*\|_2 + R^2\|\theta^*\|_\infty).$$

**Proof** Throughout the proof, $J = \{j : \theta_j^* \neq 0\}$. We proceed in three steps. Step 1 establishes initial relations and the fact that $\Delta = \hat\theta - \theta^*$ belongs to $\mathcal{C}_J(1 + \lambda + \nu)$. Step 2 provides a bound on $\frac{1}{d}\|A^\top A\Delta\|_\infty$. Finally, Step 3 establishes the rate of convergence stated in the theorem. We also often use the inequality $\|\theta\|_\infty \leq \|\theta\|_2 \leq \|\theta\|_1, \forall \theta \in \mathbb{R}^r$.

*Step 1*: We first note that,

$$\begin{aligned}
\frac{1}{d}\left\|A^\top(y - A\theta^*)\right\|_\infty &= \frac{1}{d}\|A^\top W\theta^*\|_\infty \\
&\overset{(i)}{\leq} \frac{1}{d}\left\|A^{*\top}W\theta^*\right\|_\infty + \frac{1}{d}\left\|W^\top W\theta^*\right\|_\infty \\
&\overset{(ii)}{\leq} \underbrace{\frac{1}{d}\left\|A^{*\top}W\theta^*\right\|_\infty}_{\triangleq n_1} + \underbrace{\frac{1}{d}\left\|(W^\top W - diag(W^\top W))\theta^*\right\|_\infty}_{\triangleq n_2} \\
&\qquad\qquad + \underbrace{\frac{1}{d}\left\|diag(W^\top W)\theta^*\right\|_\infty}_{\triangleq n_3},
\end{aligned}$$

where both $(i)$, $(ii)$ follow by applications of the triangle inequality. Next we bound $n_1$

$$n_1 = \frac{1}{d}\|A^{*\top}W\theta^*\|_\infty.$$

Note that the columns of $A^*$ are normalized, $\|A_i^*\|_2 = 1$ and we have $\|W\|_\infty \leq R$, thus we have all elements of $A^{*\top}W$ are bounded by $\sqrt{d}R$. We also know that $\theta^*$ is $s$-sparse, combining these we get,

$$
\begin{aligned}
n_1 &= \frac{1}{d}\|A^{*\top}W\theta^*\|_\infty \\
&\leq \frac{1}{d}(\|\theta^*\|_2)(\sqrt{s}\|A^{*\top}W\|_\infty) \\
&\leq \frac{1}{d}(\|\theta^*\|_2)(\sqrt{s}dR) \\
&\leq \|\theta^*\|_2\left(\sqrt{\frac{s}{d}}R\right),
\end{aligned}
$$

where the last step is by Cauchy-Schwartz. Next for $n_2$

$$
n_2 = \frac{1}{d}\left\|(W^\top W - diag(W^\top W))\theta^*\right\|_\infty.
$$

We know that $\|W\|_\infty \leq R$, thus we have $\|W^\top W - diag(W^\top W)\|_\infty \leq dR^2$. Again using the fact that $\theta^*$ is $s$-sparse we have,

$$
\begin{aligned}
n_2 &= \frac{1}{d}\left\|(W^\top W - diag(W^\top W))\theta^*\right\|_\infty \\
&\leq \frac{1}{d}(\|\theta^*\|_2)(\sqrt{s}\|W^\top W - diag(W^\top W)\|_\infty) \\
&\leq \frac{1}{d}(\|\theta^*\|_2)(\sqrt{s}dR^2) \\
&= \|\theta^*\|_2\sqrt{s}R^2,
\end{aligned}
$$

where the first inequality follows by an application of Cauchy-Schwartz. Finally for $n_3$, we again have $\|W^\top W\|_\infty \leq dR^2$, thus by Cauchy-Schwartz inequality

$$
n_3 = \frac{1}{d}\left\|diag(W^\top W)\theta^*\right\|_\infty \leq \|\theta^*\|_\infty R^2.
$$

Combining these together we get,

$$
\frac{1}{d}\|A^\top(y - A\theta^*)\|_\infty \leq \left(\sqrt{s}R^2 + \sqrt{\frac{s}{d}}R\right)\|\theta^*\|_2 + R^2\|\theta^*\|_\infty. \tag{12}
$$

As $\gamma = \sqrt{s}R^2 + \sqrt{\frac{s}{d}}R$, this implies that $(\theta, t, u) = (\theta^*, \|\theta^*\|_2, \|\theta^*\|_\infty)$ is feasible. Let $(\hat{\theta}, \hat{t}, \hat{u})$ be the optimal solution, then we have

$$
\|\hat{\theta}\|_1 + \lambda\|\hat{\theta}\|_2 + \nu\|\hat{\theta}\|_\infty \leq \|\hat{\theta}\|_1 + \lambda\hat{t} + \nu\hat{u} \leq \|\theta^*\|_1 + \lambda\|\theta^*\|_2 + \nu\|\theta^*\|_\infty.
$$

By rearranging terms and by triangle inequality we get the relation

$$
\|\hat{\theta}_{J^C}\|_1 \leq (1 + \lambda + \nu)\|\hat{\theta}_J - \theta^*\|_1 = (1 + \lambda + \nu)\|\Delta_J\|_1.
$$

This proves that $\Delta \in \mathcal{C}_J(1 + \lambda + \nu)$. Also by similar arguments we get

$$
\hat{t} - \|\theta^*\|_2 \leq \frac{\|\Delta\|_1 + \nu\|\Delta\|_\infty}{\lambda} \leq \frac{(1 + \nu)\|\Delta\|_1}{\lambda} \tag{13}
$$

$$
\text{and,} \quad \hat{u} - \|\theta^*\|_\infty \leq \frac{\|\Delta\|_1 + \lambda\|\Delta\|_2}{\nu} \leq \frac{(1 + \lambda)}{\nu}\|\Delta\|_1. \tag{14}
$$

*Step 2*: By applications of the triangle inequality we have

$$
\begin{aligned}
\frac{1}{d}\|A^{*\top}A^*\Delta\|_\infty &\leq \frac{1}{d}\left[\|A^\top A^*\Delta\|_\infty + \|W^\top A^*\Delta\|_\infty\right] \\
&\leq \frac{1}{d}\left[\|A^\top A\Delta\|_\infty + \|A^\top W\Delta\|_\infty + \|W^\top A^*\Delta\|_\infty\right] \\
&\leq \frac{1}{d}\left[\underbrace{\|A^\top(y - A\theta^*)\|_\infty}_{\triangleq m_1} + \underbrace{\|A^\top(y - A\hat{\theta})\|_\infty}_{\triangleq m_2} + \underbrace{\|A^\top W\Delta\|_\infty}_{\triangleq m_3} + \underbrace{\|W^\top A^*\Delta\|_\infty}_{\triangleq m_4}\right].
\end{aligned}
$$

Now we bound each of these terms

$$m_1 \overset{(i)}{\leq} d(\gamma\|\theta^*\|_2 + R^2\|\theta\|_\infty)$$

$$m_2 \overset{(ii)}{\leq} d(\gamma\hat{t} + R^2\hat{u}) \leq d\left(\gamma\|\theta^*\|_2 + R^2\|\theta^*\|_\infty + \left\{\frac{\gamma(1+\nu)}{\lambda} + \frac{R^2(1+\lambda)}{\nu}\right\}\|\Delta\|_1\right)$$

$$m_3 \overset{(iii)}{\leq} \left(dR^2 + \sqrt{d}R\right)\|\Delta\|_1 + dR^2\|\Delta\|_\infty$$

$$m_4 \overset{(iv)}{\leq} \sqrt{d}R\|\Delta\|_1,$$

where $(i)$ follows as $(\theta^*, \|\theta^*\|_2, \|\theta^*\|_\infty)$ is a feasible point, $(ii)$ is because $(\hat{\theta}, \hat{t}, \hat{u})$ is a (optimal) feasible point along with (13), (14). Bound $(iii)$ follows by similar arguments made to arrive at (12) and finally $(iv)$ is due to Hölder's inequality. Combing these we have the following bound

$$\frac{1}{d}\|A^{*\top}A^*\Delta\|_\infty \leq 2\gamma\|\theta^*\|_2 + 2R^2\|\theta^*\|_\infty + \left\{\frac{\gamma(1+\nu)}{\lambda} + \frac{R^2(1+\lambda)}{\nu}\right\}\|\Delta\|_1$$
$$+ \frac{R}{\sqrt{d}}\|\Delta\|_1 + R^2\|\Delta\|_\infty + \left(R^2 + \frac{R}{\sqrt{d}}\right)\|\Delta\|_1.$$

Simplifying using $\|\Delta\|_\infty \leq \|\Delta\|_1$ and $\gamma = \sqrt{s}\left(R^2 + \frac{R}{\sqrt{d}}\right)$ we get

$$\frac{1}{d}\|A^{*\top}A^*\Delta\|_\infty \leq 2\gamma\|\theta^*\|_2 + 2R^2\|\theta^*\|_\infty + \left(\gamma\frac{\left(1+\nu+\frac{2\lambda}{\sqrt{s}}\right)}{\lambda} + \frac{R^2(1+\lambda)}{\nu}\right)\|\Delta\|_1. \quad (15)$$

*Step 3*: Define

$$\zeta \triangleq \left(\gamma\frac{\left(1+\nu+\frac{2\lambda}{\sqrt{s}}\right)}{\lambda} + \frac{R^2(1+\lambda)}{\nu}\right).$$

Rewriting (15) using the definition of $\zeta$ we have

$$\frac{1}{d}\|A^{*\top}A^*\Delta\|_\infty \leq 2\gamma\|\theta^*\|_2 + 2R^2\|\theta^*\|_\infty + \zeta\|\Delta\|_1.$$

By the assumption on $\ell_\infty$-sensitivity and Lemma 14 we have $\kappa_1(s, u) \geq \frac{1}{2s}\kappa_\infty(s, u) \geq \frac{1}{8s}$. Thus by definition of $\ell_1$-sensitivity we have

$$\frac{1}{d}\|A^{*\top}A^*\Delta\|_\infty \geq \kappa_1(s, 1+\lambda+\nu)\|\Delta\|_1.$$

Combining this with the previous display gives us

$$\frac{1}{d}\|A^{*\top}A^*\Delta\|_\infty \leq 2\gamma\|\theta^*\|_2 + 2R^2\|\theta^*\|_\infty + \frac{\zeta}{\kappa_1(s, 1+\lambda+\nu)}\left(\frac{1}{d}\|A^{*\top}A^*\Delta\|_\infty\right)$$

$$\leq 2\gamma\|\theta^*\|_2 + 2R^2\|\theta^*\|_\infty + 8s\zeta\left(\frac{1}{d}\|A^{*\top}A^*\Delta\|_\infty\right).$$

By assumption (D5) $- 8s\zeta \leq 1/2$, therefore we have the claimed error bound

$$\frac{1}{2d}\|A^{*\top}A^*\Delta\|_\infty \leq 2\gamma\|\theta^*\|_2 + 2R^2\|\theta^*\|_\infty$$

$$\kappa_\infty(s, 1+\lambda+\nu)\|\Delta\|_\infty \leq 4\gamma\|\theta^*\|_2 + 4R^2\|\theta^*\|_\infty$$

$$\|\hat{\theta} - \theta^*\|_\infty \leq 16(\gamma\|\theta^*\|_2 + R^2\|\theta^*\|_\infty).$$

∎

# C Lower Bounds: Proof of Theorem 1

In this section we will show that when the uncertainty in the dictionary measured in matrix infinity norm scales as $R = \mathcal{O}(1/\sqrt{s})$, the $\{\ell_1, \ell_2, \ell_\infty\}$-MU Selector is information theoretically optimal up to logarithmic factors and the infinity norm of the error (in the worst case) is lower bounded by $CR\|\theta^*\|_2$. We will prove this by Fano's method (see for example review in Yu (1997); Tsybakov (2009)). The proof technique to show this estimator is minimax optimal is adapted from Belloni et al. (2016). We define the sets

$$B_0(s) = \{\theta : \|\theta\|_0 \leq s\} \qquad \text{and} \qquad S_2(L) = \{\theta : \|\theta\|_2 = L\},$$

where $L > 0$. We define the parameter set to be $\Theta = B_0(s) \cap S_2(L)$, which is the set of $s$−sparse vectors with $\|\cdot\|_2$ norm equal to $L$. To prove this theorem we will choose a particular probability distribution over the set of underlying *true dictionaries* $\mathbb{P}_{A^*}$ and also a distribution over the deviations from the true dictionary $\mathbb{P}_W$. We will assume that the entries of $A^*$ are drawn i.i.d. from a zero-mean Gaussian distribution $\mathcal{N}(0, \sigma_D^2)$ and the entries of $W$ are chosen i.i.d. from a zero mean Gaussian distribution $\mathcal{N}(0, \sigma_E^2)$ independent of the distribution generating $A^*$. We set $\sigma_D = \mathcal{O}(1/\sqrt{d})$ and $\sigma_E = \mathcal{O}(R/\sqrt{log(dr)})$. We now restate a formal version of Theorem 1.

**Theorem 17.** *Let $r \geq 2$, $2 \leq s \leq r$, and $L > 0$. Let $y = A^*\theta^*$ where $A^* \in \mathbb{R}^{d \times r}$ and $\theta^*$ is a $s$-sparse vector with norm $\|\theta^*\|_2 = L$. Further let the entries of $A^*$ be drawn from $\mathcal{N}(0, \sigma_D^2)$ and independently let the entries of the perturbation $W$ be drawn from the distribution $\mathcal{N}(0, \sigma_E^2)$. Let $A = A^* + W$, $\sigma_D^2 = \mathcal{O}(1/d)$ and $\sigma_E^2 = R/\log(dr)$. Then there exists constants $C$ and $C' > 0$ such that*

$$\inf_{\hat{T}} \sup_{\theta \in B_0(s) \cap S_2(L)} \mathbb{P}_{A^*, W} \left[ \|\hat{T} - \theta\|_\infty \geq CRL \sqrt{1 - \frac{\log(s)}{\log(r)}} \right] > C',$$

*where $\inf_{\hat{T}}$ denotes the infimum over all measurable estimators $\hat{T}$ with input $(y, A, R)$.*

**Proof** We define a finite set of "hypotheses" (packing set) included in $B_0(s) \cap S_2(L)$. To this end, we first introduce

$$\mathcal{M} = \{x \in \{0,1\}^{r-1} : \rho_H(\mathbf{0}, x) = s - 1\},$$

where $\rho_H$ denotes the Hamming distance between elements of $\{0,1\}^{r-1}$, and $\mathbf{0}$ is the zero vector. Then there exists a subset $\mathcal{M}'$ of $\mathcal{M}$ such that for any $x, x'$ in $\mathcal{M}'$ with $x \neq x'$, we have $\rho_H(x, x') > s/16$ and moreover the cardinality of $\mathcal{M}'$ is bounded below

$$\log|\mathcal{M}'| \geq Cs \log\left(\frac{r}{s}\right),$$

for some constant $C$. This follows from Varshamov-Gilbert bound (see Lemma 2.9 in Tsybakov (2009)) if $s - 1 > (r - 1)/2$ and from Lemma A.3 in Rigollet and Tsybakov (2011) if $s - 1 \leq (r - 1)/2$. We denote $\omega'_j$ to be the elements of the finite set $\mathcal{M}'$. For $j = 1, \ldots, |\mathcal{M}|$, we define the vectors $\omega_j \in \{0,1\}^r$ with components $\omega_{j1} = 0$ and $\omega_{jk} = \omega'_{j(k-1)}$ for $k > 2$, where $\omega_{jk}$ is the $k$-th component of $\omega_j$. We also define $\omega_0$ as the vector in $\{0,1\}^r$ with all components equal to 0 except the first one equal to 1. We now define the set of "hypotheses" (packing set of $\Theta$) $(\bar{\omega}_j, j = 0, \ldots, |\mathcal{M}'| + 1)$, where $\bar{\omega}_0 = R\omega_0$ and

$$\bar{\omega}_j = \frac{L}{\sqrt{1 + \psi^2(s-1)}}(\omega_0 + \psi\omega_j), \qquad j = 1, \ldots, |\mathcal{M}'| + 1.$$

Here $\psi$ is a positive parameter that will be chosen appropriately. Note that these vectors are $s$-sparse and have $\|\bar{\omega}_j\|_2 = L$. By Lemma 18 we have the KL divergence is bounded,

$$\begin{aligned}
\mathcal{K}(\mathbb{P}_{\bar{\omega}_j}, \mathbb{P}_{\bar{\omega}_0}) &= \frac{d\sigma_D^2}{2\sigma_E^2 \|\bar{\omega}_0\|_2^2} \|\bar{\omega}_j - \bar{\omega}_0\|^2 \\
&\leq \frac{d\sigma_D^2}{2\sigma_E^2 L^2} \left(\frac{\psi^2 L^2 s}{1 + \psi^2(s-1)}\right) \\
&\leq \psi^2 \left(\frac{s d\sigma_D^2}{2\sigma_E^2(1 + \psi^2(s-1))}\right).
\end{aligned}$$

If we choose $\psi = C\sqrt{\frac{\sigma_E^2 \log(r/s)}{d\sigma_D^2}}$ with $C$ being an appropriately chosen constant independent of dimensions $(s, d, r)$ and $L$ we get that for all $j$,

$$\mathcal{K}(\mathbb{P}_{\bar{\omega}_j}, \mathbb{P}_{\bar{\omega}_0}) \leq \frac{1}{16} \log(|\mathcal{M}'|).$$

Thus for $j$ and $j'$ both different from 0,

$$\|\bar{\omega}_j - \bar{\omega}_{j'}\|_\infty = \frac{L\psi}{\sqrt{1 + \psi^2(s-1)}} \geq C\frac{L\sigma_E\sqrt{\log(r/s)}}{\sqrt{d}\sigma_D},$$

and for $j \neq 0$ we have

$$\|\bar{\omega}_j - \bar{\omega}_0\|_\infty \geq \frac{L\psi\|\omega_j\|_\infty}{\sqrt{1 + \psi^2(s-1)}} \geq C\frac{L\sigma_E\sqrt{\log(r/s)}}{\sqrt{d}\sigma_D}.$$

We want the columns of $\|A^*\|_2 \leq 1$ (upper bound used in the proof of Theorem 16), hence we want $\sigma_D = \mathcal{O}(1/\sqrt{d})$ (this follows by an application of Lemma 13 followed by a union bound over the $r$ columns using the fact that $r = \mathcal{O}(poly(d))$). We also demand that our deviation from the true dictionary be bounded by $R$ with high probability over all entries so we choose $\sigma_E \leq \mathcal{O}(R/\sqrt{log(dr)})$. Hence given our choices of $\sigma_E$ and $\sigma_D$ we have for any $j, j'$

$$\|\bar{\omega}_j - \bar{\omega}_{j'}\|_\infty \geq CLR\left(\sqrt{1 - \frac{\log(s)}{\log(r)}}\right).$$

We can now apply Theorem 2.7 in Tsybakov (2009) to complete the proof. ∎

**Lemma 18.** *Let $\theta_1 \in \mathbb{R}^r$ and $\theta_2 \in \mathbb{R}^r$ be such that $\|\theta_1\|_2 = \|\theta_2\|_2$. Under the assumptions stated in the Appendix C we have*

$$\mathcal{K}(\mathbb{P}_{\theta_1}, \mathbb{P}_{\theta_2}) = \frac{d\sigma_D^2}{2\sigma_E^2\|\theta_2\|_2^2}\|\theta_1 - \theta_2\|^2.$$

**Proof** By the properties of Kullback Leibler divergence between product measures, it suffices to prove the lemma for $d = 1$. Let $\theta \in \mathbb{R}^r$. Consider the random vector $(U, V)$ where

$$V = (D_1 + E_1, \ldots, D_r + E_r),$$

with $D = (D_1, D_2, \ldots, D_r)^\top$ a zero-mean Gaussian vector with covariance $\sigma_D^2 I_{r\times r}$ and $E = (E_1, E_2, \ldots, E_r)^\top$ a zero mean Gaussian vector with covariance $\sigma_E^2 I_{r\times r}$ independent of A and

$$U = \sum_{j=1}^r \theta_j(V_j - E_j).$$

We introduce some variables

$$\tilde{\Sigma} = \frac{\sigma_E^2}{\sigma_D^2 + \sigma_E^2}I_{r\times r}, \quad \Pi = \frac{\sigma_D^2}{\sigma_D^2 + \sigma_E^2}I_{r\times r}, \quad c_\theta = \theta^\top\Pi\theta = \frac{\sigma_D^2}{\sigma_D^2 + \sigma_E^2}\|\theta\|_2^2.$$

We find the conditional distribution $\mathcal{L}_\theta(U|V)$ of $U$ given $V$. Also note that the vector $(V_1, \ldots, V_r, E_1, \ldots, E_r)^\top$ is a zero-mean Gaussian random vector with covariance matrix

$$\begin{pmatrix} (\sigma_D^2 + \sigma_E^2)I_{r\times r} & \sigma_E^2 I_{r\times r} \\ \sigma_E^2 I_{r\times r} & \sigma_E^2 I_{r\times r} \end{pmatrix}.$$

So that $\mathcal{L}_\theta(E|V)$ is a Gaussian with mean $\tilde{\Sigma}V$ and covariance $\sigma_E^2(I_{r\times r} - \tilde{\Sigma})$. This implies that $\mathcal{L}_\theta(U|V)$ is Gaussian with mean $\theta^\top\Pi\theta$ and variance $c_\theta\sigma_E^2$. Then the logarithm of density of $\mathcal{L}_\theta(U|V)$, denoted by $\ell_\theta(U|V)$ satisfies

$$\ell_\theta(U|V) = -\frac{1}{2}\log(2\pi) - \frac{1}{2}\log(c_\theta\sigma_E^2) - \frac{1}{2c_\theta\sigma_E^2}(U - \theta^\top\Pi V)^2.$$

Now let $\theta_1 \in \mathbb{R}^r$ and $\theta_2 \in \mathbb{R}^r$ with $\|\theta_1\|_2 = \|\theta_2\|_2$. Then,

$$\ell_{\theta_1}(U|V) - \ell_{\theta_2}(U|V) = \frac{1}{2}\underbrace{\left(\log\left(\frac{c_{\theta_2}}{c_{\theta_1}}\right)\right)}_{=0} + \frac{1}{2c_{\theta_2}\sigma_E^2}\left((U - \theta_2^\top \Pi V)^2 - (U - \theta_1^\top \Pi V)^2\right)$$

$$+ \underbrace{\left(\frac{1}{2c_{\theta_2}\sigma_E^2} - \frac{1}{2c_{\theta_1}\sigma_E^2}\right)}_{=0}(U - \theta_1^\top \Pi V)^2$$

$$= \frac{1}{2c_{\theta_2}\sigma_E^2}\left((U - \theta_2^\top \Pi V)^2 - (U - \theta_1^\top \Pi V)^2\right).$$

Since the distribution of $V$ does not depend on $\theta_1$ we obtain that in the case $d = 1$,

$$\mathcal{K}(\mathbb{P}_{\theta_1}, \mathbb{P}_{\theta_2}) = \frac{1}{2c_{\theta_2}\sigma_E^2}\mathbb{E}_{\theta_1}\left[(U - \theta_2^\top \Pi V)^2 - (U - \theta_1^\top \Pi V)^2\right]$$

$$= \frac{\sigma_D^2 + \sigma_E^2}{2\sigma_E^2\sigma_D^2\|\theta\|_2^2}\Big[\sigma_D^2(\theta_1^\top - \theta_2^\top \Pi)I_{r\times r}(\theta_1 - \Pi\theta_2)$$

$$- \sigma_D^2(\theta_1^\top - \theta_1^\top \Pi)I_{r\times r}(\theta_1 - \Pi\theta_1) + (\theta_2^\top \Pi^2\theta_2 - \theta_1^\top \Pi^2\theta_1)\Big].$$

Where in the final step the cross terms are zero by the independence of $D$ and $E$. Developing this expression leaves us with

$$\mathcal{K}(\mathbb{P}_{\theta_1}, \mathbb{P}_{\theta_2}) = \frac{\sigma_D^2 + \sigma_E^2}{2\sigma_E^2\sigma_D^2\|\theta\|_2^2}\left[(\theta_1 - \theta_2)^\top \Pi\sigma_D^2 I_{r\times r}(\theta_1 - \theta_2)\right]$$

$$= \frac{\sigma_D^2}{2\sigma_E^2\|\theta_2\|_2^2}\|\theta_1 - \theta_2\|^2.$$

$\blacksquare$