[Reviews · NeurIPS 2017]

Reviewer 1



This paper proposes and analyzes an alternating minimization-based algorithm to recover the dictionary matrix and sparse coefficient matrix in a dictionary learning setting. A primary component of the contribution here comes in the form of an alternate analysis of the matrix uncertainty (MU) selector of Belloni, Rosenbaum, and Tsybakov, to account for worst-case rather than probabilistic corruptions. Pros: + The flavor of the contribution here seems to improve (i.e., relax) the conditions under which methods like this will succeed, relative to existing works. Specifically, the motivation and result of this work amounts to specifying sufficient conditions on the vectorized infinity norm of the unknown dictionary matrix, rather than its operator norm, under which provable recovery is possible. This has the effect of making the method potentially less dependent on ambient dimensions, especially for "typical" constructions of the (incoherent) dictionaries such as certain random generations. + The alternate analysis of the MU selector is independently interesting. Cons: - It would be interesting to see some experimental validation of the proposed method, especially ones that investigate the claimed improvements in the conditions on the unknown dictionary relative to prior efforts. In other words, do the other efforts that state results in terms of the operator norm fail in settings where this method succeeds? Or are the methods all viable, but just a more refined analysis here? This is hard to determine here, and should be explored a bit, I think. - The paper is hard to digest, partly because of notation, and partly because of some pervasive grammatical and formatting issues: - Line 8 of algorithm 1, as written, seems to require knowledge of the true A,x quantities to compute. In reality, it seems this should be related somehow to the samples {y} themselves. Can this be written a bit more clearly? - Condition (c4) on page 6, line 206 is confusing as written. The dimension of x is r, and it is s-sparse, so there are more than r options for *sets* of size s; this should be r-choose-s, I guess. The subsequent conditions are apparently based on this kind of model, and seem to be correct. - Why include the under brace in the first equation of line 251 on page 7? Also, repeating the LHS is a little non-standard. - The "infinite samples" analysis is a little strange to me, too. Why not simply present and analyze the algorithm (in the main body of the paper) in terms of the finite sample case? The infinite case seems to be an analytical intermediate step, not a main contribution in itself. - There are many sentence fragments that are hard to parse, e.g., "Whereas..." on line 36 page 2, "While..." on line 77 page 2, and "Given..." on line 131 page 4.

Reviewer 2



This paper give theoretical guarantees for an alternating minimization algorithm for the dictionary learning and sparse coding problem. The analysis is based on the infinity norm of the dictionary and gives linear convergence rate for sparsity level s = O(sqrt(d)). The sparse regression estimator is the {l_1, l_2, l_\infty}-MU selector and the dictionary learning step is an one-step steepest descent method. The idea of the alternating minimization algorithm is not new. The new aspect of the algorithm in this paper is that {l_1, l_2, l_\infty}-MU selector is used for sparse regression. The main interesting result in this paper is its theoretical analysis. It uses infinity norm instead of 2 norm for the dictionary and gives a better sparsity level when compared to the best previous results in the overcomplete setting. However, this paper is not easy to follow for non-experts. The authors suppose readers have sufficient background in this areas and do not give a complete introduction. See comments below. *), The title claims the proposed method is fast. However, no experiments are given in this paper. Since this new algorithm has a better bound of sparsity level theoretically, it also would be good to have numerical comparisons to show the probability of successful recovery versus sparsity level. *), P3, L88: Algorithm 1 is not clear at the current stage. For example, y is in R^d from L18. Therefore, by L77 in paragraph of notation, y_k in Step 3 is supposed to be a scalar. However, from L110 (2), y_k should be a vector in R^d. I suppose the samples should be a matrix Y and each column is denoted by y_k \in R^d. \mathbb{I} in Step 5 is used without introduction. The motivation of Step 8 is not given. It is a steepest descent update, right? If yes, the cost function should be given. A^* is used in Step 8, but A^* is unknown. The authors could add a intermediate step which use the given samples y instead of A^*. In summary, Algorithm 1 needs be clarified. *), P7, L238. What cost function is the gradient \hat{g}_{i j} with respect to?

Reviewer 3



The paper proposes an alternating minimization algorithm for dictionary learning, and theoretical guarantees are also given. In each step the algorithm first uses an l1, l2 and l_infty algorithm with thresholding to get an estimate of the coefficients, and then use another gradient step to update the dictionary. To me two shining points of the paper: 1. Guarantee holds for the overcomplete dictionary. 2. Improved the sparsity level requirement by a factor of log d. Obviously the NIPS format is too short for the arguments the authors are making, and a lot of details are moved to the appendix. Due to time limit I cannot read all the details of the proof. Below are some questions: 1. In A1 you have a mu-incoherence assumption, but mu is not shown in your theorem 3. Is it hidden somewhere? 2. In assumption B1 you mentioned, and I agree that there is a fast random initialization so that the condition holds. Can you give some details about your initialization procedure and guarantees? 3. How do you handle the permutation invariance of A? 4. In your algorithm 1, line 3, the MUS algorithm has a return, but in your definition (equation 2), the return is not specified. Actually the returned should be theta instead of (theta, t, u). 5. In your algorithm 1, line 3, can you give some explanation about “(w_k^t is the k^th covariate at step t)”? Why w_k^t is called the k^th covariate? 6. Any simulation result verifying your convergence rate?